# Age Estimation in *Sepia officinalis* Using Beaks and Statoliths

**DOI:** 10.3390/ani14152230

**Published:** 2024-07-31

**Authors:** Blondine Agus, Stefano Ruiu, Jacopo Cera, Andrea Bellodi, Viviana Pasquini, Danila Cuccu

**Affiliations:** 1Department of Life and Environmental Science, University of Cagliari, 09126 Cagliari, Italy; blondine.agus@unica.it (B.A.); stefanoruiu96@gmail.com (S.R.); jacopo.cera96@gmail.com (J.C.); abellodi@unica.it (A.B.); viviana.pasquini@unica.it (V.P.); 2Stazione Zoologica (SZN) Anton Dohrn, Contrada Porticatello 29, 98167 Messina, Italy

**Keywords:** *Sepia officinalis*, age, beaks, statoliths, Mediterranean

## Abstract

**Simple Summary:**

Determining the age of cephalopods is essential for understanding their life history, which is in turn crucial for assessment and management. The short lifespan with a single reproductive event, high natural mortality rates, and rapid growth in these species make the application of traditional age-based models impractical. Most hard structures of cephalopods, such as mandible (beaks) and statoliths, can store ontogenetic events through the formation of periodic marks or growth increments. Statoliths are paired calcareous structures located in two anterior chambers of the head. In this work, for the first time in the Mediterranean, we successfully performed age estimation in wild cuttlefish *Sepia officinalis* using beaks and statoliths. Based on daily increments that were previously validated in statoliths, the beaks were cross-verified. As the beak presented more advantages in age studies than statoliths, due to the relative simplicity of its processing method, it was eventually proposed as a suitable hard structure to study the age of *S. officinalis*.

**Abstract:**

Establishing the age of cephalopods is crucial for understanding their life history, which can then be used for assessment and management. This is particularly true for the common cuttlefish *Sepia officinalis* (Linnaeus, 1758), one of the most important resources for coastal fisheries. For this reason, an age analysis of *S. officinalis* was performed for the first time in the Mediterranean, using beaks and statoliths from 158 wild specimens (55–222 mm mantle length; 23–1382 g total weight) at different maturity stages (immature to mature). Growth increments were counted in the lateral wall of the upper beaks and the lateral dome of statoliths. In both cases, a good relationship was found between the counts and the sizes of the animals. The low values of coefficients of variation between the readings obtained for beaks (3.96 ± 1.87%) and statoliths (4.00 ± 1.89%) showed a high level of precision and accuracy in the readings. However, the analysis was simpler for beaks, which were all successfully analyzed, while it was more complex for statoliths, with 69% being lost due to rejection or overgrinding. Based on daily increments previously validated in statoliths, the beaks were cross-verified by comparing their counts with those from statoliths extracted from the same 83 specimens, obtaining a statistically significant relationship between the two counts, confirmed by the ANOVA test. Absolute growth rates that were assessed using both beaks and statoliths indicated that the two sexes had a higher growth rate at 122 and 182 days, which subsequently declined in older specimens. Due to the relative simplicity of its processing method, the beak was finally proposed as a suitable hard structure to study the age of *S. officinalis*. We also confirmed the good readability of increments in the lateral wall of the beak, which could be considered a valid alternative to the rostrum surface.

## 1. Introduction

The common cuttlefish *Sepia officinalis* (Linnaeus, 1758) is one of the most important coastal fishery resources in both the Eastern Atlantic [1] and the Mediterranean Sea [2,3,4,5,6]. In most Mediterranean areas, it is exploited at both industrial and small-scale levels, including in recreational angling [7]. This species, together with *Octopus vulgaris*, *Eledone moschata*, and *E. cirrhosa*, accounts for 15% of the total landing value [8]. Despite its economic importance, studies assessing fisheries and the stock status of *S. officinalis* in the Mediterranean are rather scarce [9]. In general, the age determination of cephalopods is critical to understanding their life history and modeling their population dynamics, which are in turn essential for assessment and management.

However, in coleoid cephalopods, the short lifespans; semelparous reproduction; high natural mortality rates; and rapid, often non-asymptotic growth make the use of traditional age-based models impractical [10]. Since the observation of growth lines on *Moroteuthis ingens* beaks [11], hard structures in cephalopods have been increasingly considered to evaluate the presence of increments and their deposition process. 

Age estimates for *Octopus vulgaris* have been successfully made by counting growth increments on the upper beak’s rostrum sections [12] and in the lateral walls [13].

Daily deposition was validated for *O. vulgaris* [14], and since then, beaks have been used for studying aging in several species [15,16,17]. Techniques using statoliths, despite being laborious and time-consuming, are the most common methods used for studying aging in cephalopods, and their reliability has been proven in several species [18,19,20]. Other hard structures, such as stylets, gladius, and eye lenses, have also been investigated to determine their usability in age estimation [21,22,23,24]. Age studies on *S. officinalis* have been carried out mostly using statoliths, by reading the sequence of increments in the lateral dome [25,26,27]. In this structure, a daily periodicity of the increments was validated on individuals less than 240 days old, but in older specimens, the age was underestimated due to the poor resolution of the later growth increments [25]. Additionally, the internal shell (cuttlebone) of *S. officinalis* was utilized to examine age; however, no relationship was found between the number of counted lamellae and the actual age of the animals [25]. More recently, in this species, the beak has been taken into account for aging investigations considering adult sizes [28,29] and early stages [30] using the rostrum surface, but the daily deposition of increments has been validated only in the latter study. Currently, there is no study related to the age estimation of the common cuttlefish in the Mediterranean. The present paper aims to perform age estimation in specimens of *Sepia officinalis* caught in the Mediterranean Sea using statoliths and beaks to test the reliability of these structures and suggest the best one suitable for age study in this species.

## 2. Materials and Methods

### 2.1. Sample and the Processed Hard Structures 

A sample of 158 *S. officinalis* was caught dead between October 2022 and May 2023 from the artisanal fishery (trammel nets and pots) in the western Mediterranean Sea (Sardinian Seas) within an experimental project. For each specimen, the dorsal mantle length (ML; mm) and the total weight (TW; g) were recorded, the sex and sexual maturity were determined [31], and the beaks and statoliths were extracted.

After extraction, all beaks were cleaned and preserved in 70% ethyl alcohol [13]. The upper beak was selected for age estimation analysis. Each upper beak was sagittally sectioned to obtain two symmetrical sections. One of them was cleaned with water and analyzed under a stereoscopic microscope (Stemi 2000-C, Zeiss, Oberkochen, Germany) equipped with transmitted light (SZ-ST8 OPTIKA Microscopes, Ponteranica, Italy). Each extracted statolith was treated according to [32], rinsing the structure in distilled water and drying it. It was then mounted on a microscope slide using Crystalbond™ 509 (Agar Scientific Ltd., Stansted, UK) and then ground on both sides using 30, 12, 5, and 0.5 μm grit waterproof sandpaper.

### 2.2. Increment Counting and Statistical Analysis

Growth increments in beaks and statoliths were counted by image analysis using Tps_Dig2, 2.32 software.

The images of the beak increments were taken using a stereo microscope (Stemi 2000-C, Zeiss) equipped with a camera. The number of growth increments (NIs) of each upper beak was counted from the rostral tip area to the opposite end of the lateral wall [13]. In the case of worn parts (scratches), the incomplete increments were counted regardless, continuing in the region not damaged to avoid underestimation. For each statolith, using an optical microscope equipped with a CANON EOS 1100 D camera (Canon Italia, Milano, Italy)(magnifications: 100× and 400×), the total number of growth increments (NIs) was counted from the natal ring toward the edge of the lateral dome, considering the periodicity of the increments (daily) based on studies on other sepiid species [19,25]. For both structures, the counting was performed twice by the same trained reader at different times. The mean value obtained from the readings designated the estimated age.

The overall readings’ precision and accuracy were evaluated by the coefficient of variation (%CV) [33], and the mean number of increments of the readings was considered [14,30]. The relationships between NI, ML, and TW were described separately for both sexes with four different curve models: linear, exponential, power, and logarithmic. Afterward, the Akaike Information Criterion (AIC) was calculated to choose which model best described growth. The absolute growth rate (AGR; mm d^−1^) was calculated separately for beaks and statoliths for each 60-day interval by sex using the following equation [34]:AGR = R_2_ − R_1_/T_2_ − T_1_
where R_1_ and R_2_ are the average ML (mm) at the beginning (T_1_) and end (T_2_) of the time interval, respectively.

In 83 individuals (juveniles and adults), the increments counted in both the statoliths and in beaks were compared using linear regression and ANOVA test analysis with the software Statgraphics Centurion XVI (16.1.11 version).

## 3. Results

### Sample and the Hard Structures 

The total sample comprised 86 males and 72 females at different maturity stages (from immature to mature). The minimum sexual maturity size was observed at 90 mm ML (94 g TW) in males and 106 mm (118 g) in females. All the extracted upper beaks showed a distinct sequence of increments, clearly visible for counting (Figure 1), including small, partially pigmented beaks belonging to juvenile animals (Figure 1A,B). In no case was the increment visibility on the anterior region near the rostrum poor. Scratches that only partially obscured the visibility of the increments were observed in some peripheral regions of the lateral wall of a few beaks (N = 5) belonging to the largest individuals (ML ≥ 200 mm) (e.g., Figure 1C).

The lowest number of increments (N = 53) was counted on the beaks of a female (59 mm; 29.03 g) and a male (55 mm; 23.93 g), both sexually immature.

The highest values of increments, 332 and 310, were observed in the largest animals that were fully mature: a female (220 mm; 1050 g mature) and a male (222 mm; 910.7 g mature). As a result of age estimation, sexual maturity was observed at a minimum age of 96 days and 118 days in males and females, respectively. Overall, our mature samples had an average age of 177 ± 50 days in males and 189 ± 52 days in females (Table 1).

The low values of CV (3.96 ± 1.87%) showed a high level of precision and accuracy in the readings. The AIC test (Appendix A) showed that the model that best described both the NI-ML and NI-TW relationships for both sexes was the power model (Figure 2A,B).

As shown in Table 2, in both sexes, absolute growth rates were initially higher and then gradually decreased. The highest values were found in the age class 122–182 days, with females growing faster (0.72 mm/day) than males (0.53 mm/day).

Growth increments were observed in the statoliths (Figure 3A,B), but the processing was difficult, resulting in many statoliths being rejected or excessively ground (69%). The low values of CV (4.00 ± 1.89) showed a high level of precision and accuracy in the readings.

The number of increments counted in the statoliths ranged from 60 to 205 in females and between 66 and 214 in males (Table 3). The AIC test (Appendix A) showed that the model that best described both the NI-ML and NI-TW relationships for both sexes was the exponential model (Figure 4A,B).

The AGR values obtained by statolith age showed the same trend observed in beaks, with the highest values in the age class 122–182 days for females (0.93 mm/d) and males (0.85 mm/d) (Table 4).

A good relationship was found between the increments counted in the beaks and statoliths from the same 83 specimens (Figure 5). The correlation coefficient of 0.909416 indicates a relatively strong relationship between the variables, and the *p*-value < 0.05 of the ANOVA test confirms a statistically significant relationship at the 95.0% confidence level between the number of increments. However, it should be noted that the relationship changed when the number of increments was higher than 150; indeed, the number of increments counted in beaks was always higher (e.g., 332 vs. 214) (Figure 5). 

## 4. Discussion

This is the first age estimation of *Sepia officinalis* from the Mediterranean using hard structures across a wide range of sizes and different sexual maturity stages (immature, developing/maturing, and mature). Sexual maturation was observed from 90 mm ML earlier in males, as already reported in other areas of the Mediterranean Sea [35,36]. Focusing on age, a preliminary study conducted in the Central-East Atlantic showed that beaks could be a promising tool for age estimation in *Sepia officinalis* using the rostral section because the visibility of increments was better than that of lateral walls in the whole known-age sample (from 1 to 175 days old) [28]. Another recent study in the same geographical area on *S. bertheloti* selected the rostrum section because it showed the clearest pattern of growth increments [37]. Our study on beaks returned a good outcome given the perfect visualization of the growth increments along the lateral wall. The increments were always clearly visible, allowing us to determine their counts in beaks belonging to animals of both sexes and at different levels of sexual maturation. The literature suggests that age estimation using beaks from large individuals may lead to underestimation due to the erosion of the rostrum tip, which is the part more exposed during predation [20]. In this case, counting the oldest increments in the dorsal area of the section [14,20] or using an imaging system to measure increment width and extrapolate the increment number of the eroded portion prevents age underestimation [38]. In our samples, the increment visibility in the anterior region near the rostrum was never poor, and no extrapolation was necessary. In *S. officinalis*, scratches are occasionally present in some parts of the beak, as well as the lateral wall, due to mechanical damage to the chitinous tissues of the beak during feeding [39]. Indeed, in a few of the larger animals (above 200 mm ML), we observed scratches in small peripheral regions of the lateral wall that partially obscured the visibility of the increments but did not prevent their reliable sequence from being observed. Overall, the reliability of the data obtained from the lateral wall of the upper beaks is strengthened by the excellent degree of correlation between the number of increments counted and the size of the animals for both sexes, as well as by the low CV% values that highlight the precision of the reading (no beaks were discarded). Regarding statoliths, even though the low CV% highlighted precise and reliable readings, the greatest difficulty was in the processing and consequently in the high loss of rejected or overgrinded structures, which is common in cephalopod aging studies [25]. For the statoliths of *S. officinalis*, the hypothesis of 1 increment = 1 day in individuals younger than 240 days has been assumed [25]. Assuming this, in the present analysis, upper beaks were cross-verified by comparing their counts with those from statoliths extracted from the same individuals (juveniles and adults), obtaining a statistically significant relationship between the two counts, as confirmed by the ANOVA test. It should be noted that, for larger mature animals, the number of increments counted in beaks was higher than that in statoliths. Under natural conditions, *S. officinalis* reaches sexual maturity in a wide range of sizes and has a life cycle of 12–24 months. The biological life cycle in the Mediterranean lasts 1 year, and specimens lay eggs at smaller mature sizes (Group I breeders), while in the Atlantic, they reach maturity at larger mature sizes and breed at approximately 20 months old (Group II breeders) [5,40]. This characteristic is supposed to be the consequence of water temperature and/or food availability and the hatching date [5], which allow the individuals to attain a threshold size at sexual maturity with a higher initial growth rate in the first year or extend their life cycle with a lower initial growth rate to the second year of breeding [40]. The AGR values obtained in this work using both beaks and statoliths, even if preliminary, confirm a higher initial growth rate for both sexes, which gradually declines. The highest values observed between 122 and 182 days, when the individuals were not yet mature, suggest that before reaching sexual maturity, individuals consume energy for growth.

Agreeing that the Mediterranean population has a 1-year life cycle, for our larger mature specimens (close to death), in which the number of increments counted in beaks was higher than that in statoliths (e.g., 332 vs. 214), we are inclined to consider the beak count to be more realistic than those obtained from statoliths, which may lead to underestimation. This is in line with what was already observed in older Atlantic specimens (around 240 days) [25,41]. The underestimation, as already explained by other authors, could be linked to the missing of very narrow increments during the count or to the high statolith calcification and, thus, poor increment definition [41].

## 5. Conclusions

In conclusion, considering all results and agreeing that age estimation methods must be accurate, precise, and not laborious or very time-consuming [42], we also agree that beaks present greater advantages than statoliths in age studies due to the relative simplicity of their processing method [15]. For these reasons, we recommend the beak as a suitable hard structure to study the age of cuttlefish *S. officinalis*. We also confirmed the good readability of increments in the lateral wall of the beak, which could be considered a valid alternative to the rostrum surface.

## Figures and Tables

**Figure 1 animals-14-02230-f001:**
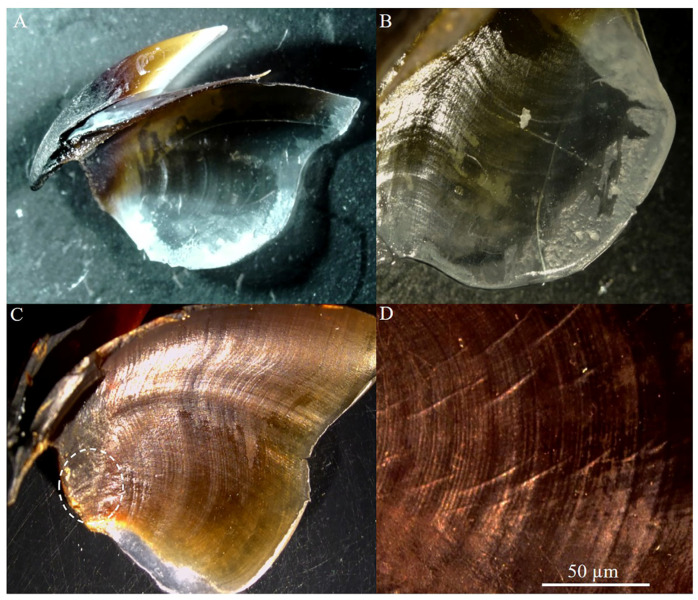
Upper beaks (**A**,**C**) and details of increments in the lateral wall (**B**,**D**) of an immature male (55 mm ML; 23.93 g TW) and a mature female (213 mm ML; 1048.7 g TW) of *Sepia officinalis*, respectively.

**Figure 2 animals-14-02230-f002:**
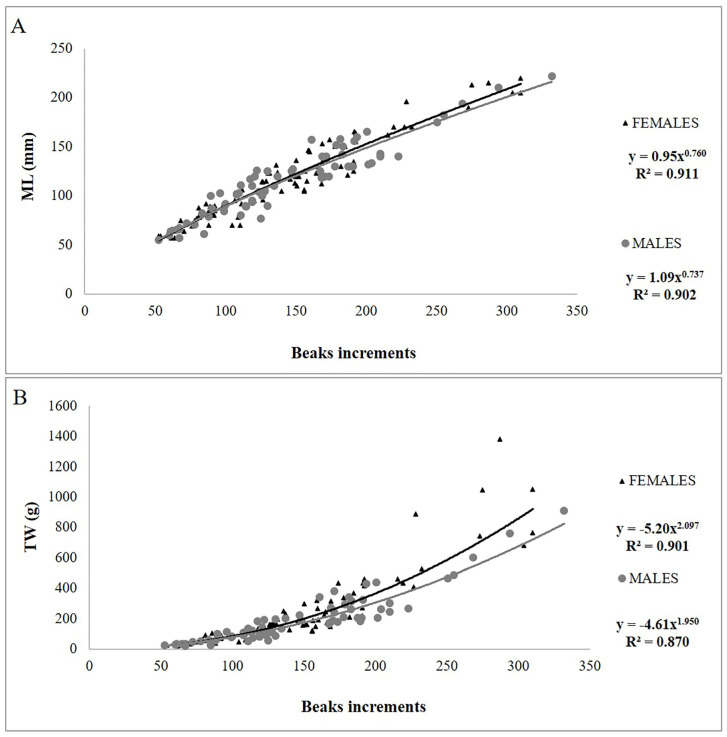
Plot of the number of increments counted in the upper beak vs. dorsal mantle length (ML) (**A**) and total weight (TW) (**B**) for males and females of *Sepia officinalis*.

**Figure 3 animals-14-02230-f003:**
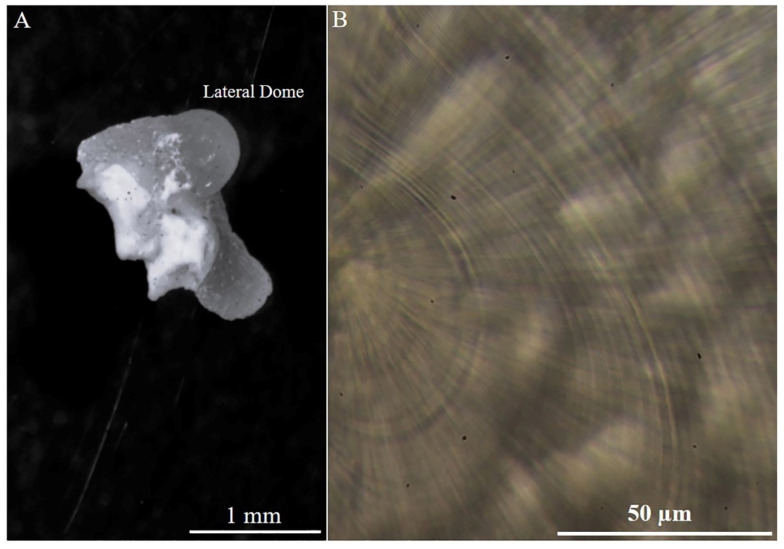
(**A**) Statoliths of a mature male (143 mm ML; 301.8 g TW) of *Sepia officinalis* and (**B**) details of increments counted in the lateral dome region (magnification 400×).

**Figure 4 animals-14-02230-f004:**
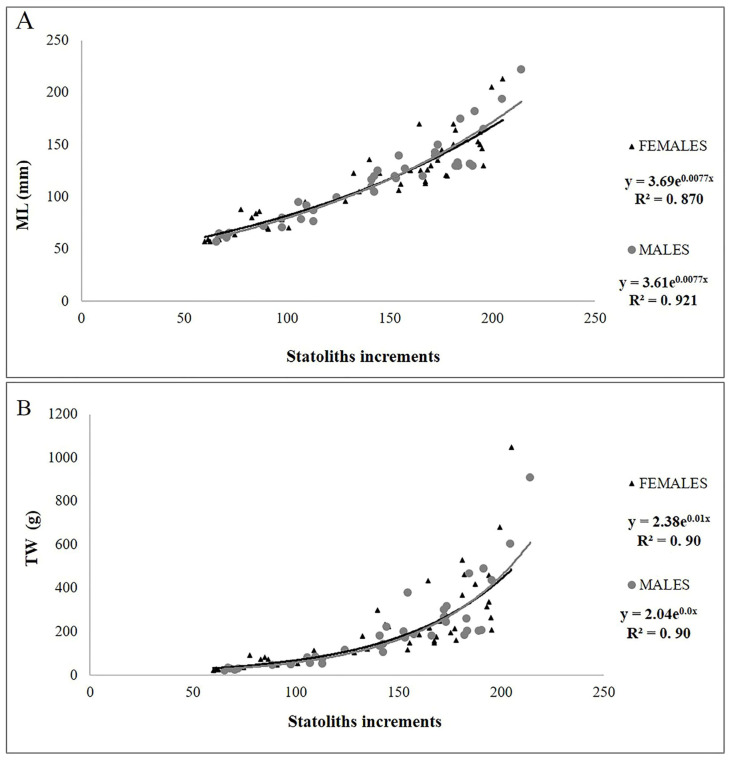
Plot of the number of increments counted in the statoliths vs. dorsal mantle length (ML) (**A**) and total weight (TW) (**B**) for males and females of *Sepia officinalis*.

**Figure 5 animals-14-02230-f005:**
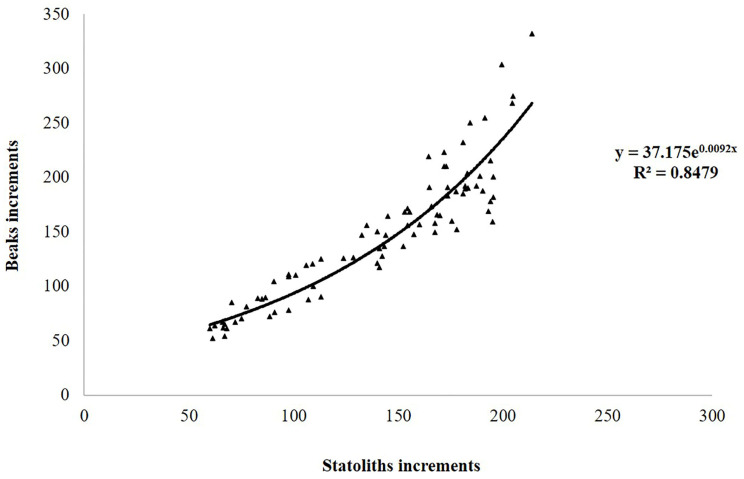
Relationship between the increments counted in beaks and statoliths from the same specimens of *Sepia officinalis*.

**Table 1 animals-14-02230-t001:** Details of the sample of *Sepia officinalis* and number of increments (NIs) counted in beaks of males and females at different sexual maturity stages, with mean and standard deviation in parenthesis.

Sex	Maturity Stage	N	ML (mm)	TW (g)	NI
					Beak
Males N 72	Immature	9	55–71 (62.88 ± 5.08)	23.6–51.4 (32.37 ± 8.31)	53–85 (67 ± 9)
	Developing and maturing	20	72–110 (91.25 ± 15.51)	48,73–112.15 (81.52 ± 40.67)	73–130 (106 ± 16)
	Mature	43	90–222 (137.48 ± 28.25)	93.78–910.70 (276.06 ± 169.19)	96–332 (177 ± 50)
Females N 86	Immature	12	57–75 (65.50 ± 6.50)	22,45–48,73 (36.77 ± 8.92	53–105 (70 ± 15)
	Developing and maturing	33	70–126 (96.93 ± 16.61)	49.23–232.70 (112.58 ± 47.79)	80–169 (114 ± 27)
	Mature	41	106–220 (148.34 ± 32.48)	118.40–1382.07 (389.56 ± 287.91)	118–310 (189 ± 52)

**Table 2 animals-14-02230-t002:** Absolute growth rates (AGRs) for mantle length for age class of increments in beaks of *Sepia officinalis* by sex. $\bar{x}$, average; SD, standard deviation.

Sex	Age Class (days)	N	ML $\bar{x}$ ± SD	AGR (mm/d)
Males N 72	<60	2	57.50 ± 3.5	-
	61–121	30	88.23 ± 17.20	0.51
	122–182	22	123.05 ± 20.41	0.53
	183–243	13	146.81 ± 12.07	0.44
	>244	5	170.38 ± 29.40	0.38
Females N 86	<60	2	59.00 ± 0	-
	61–121	31	80.32 ± 12.83	0.36
	122–182	35	123.66 ± 14.93	0.72
	183–243	12	156.92 ± 21.37	0.55
	>244	6	191.67 ± 14.50	0.48

**Table 3 animals-14-02230-t003:** Details of the sample of *Sepia officinalis* and number of increments (NIs) counted in statoliths of males and females at different sexual maturity stages, with mean and standard deviation in parenthesis.

Sex	Maturity Stage	N	ML (mm)	TW (g)	NI
					Statoliths
Males N 38	Immature	6	57–71 (64 ± 4.73)	23. 6–51.4 (33.85 ± 9.58)	66–98 (73 ± 12)
	Developing and maturing	7	72–95 (83.14 ± 8.39)	48.73–82.1 (65.57 ± 15.32)	89–113 (105 ± 9)
	Mature	25	100–222 (138.72 ± 28.91)	109.17–910.7 (286.24 ± 181.05)	124–214 (169 ± 23)
Females N 45	Immature	8	57–70 (62.6 ±5.31)	22.45–48.73 (33.92 ± 9.49)	60–91 (72 ± 13)
	Developing and maturing	13	70–126 (98.07 ± 19.29)	54–232.7 (119.65 ± 55.01)	78–169 (118 ± 33)
	Mature	24	105–213 (143.91 ± 27.54)	118.4–1048.7 (331.21 ± 207.4)	135–205 (176 ± 19)

**Table 4 animals-14-02230-t004:** Absolute growth rates (AGRs) for mantle length for age class of increments in statoliths of *Sepia officinalis* by sex. $\bar{x}$, average; SD, standard deviation.

Sex	Age Class (Days)	N	ML $\bar{x}$ ± SD	AGR (mm/d)
Males N 38	<60	-	-	-
	61–121	13	74.31 ± 11.96	1.24
	122–182	16	125.32 ± 14.35	0.85
	183–243	9	162.55 ± 33.51	0.50
Females N 45	<60	1	57.00 ± 0	-
	61–121	14	73.21 ± 12.03	0.27
	122–182	22	128.82 ± 20.22	0.93
	183–243	8	164.25 ± 29.18	0.47

## Data Availability

The data presented in this study are available from the corresponding author upon reasonable request.

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
