# Peer review of "Age Estimation in Sepia officinalis Using Beaks and Statoliths"

_animals, 2024, doi:10.3390/ani14152230_

Round 1
Reviewer 1 Report
Comments and Suggestions for Authors
This paper was described the aging of Sepia officinalis in the Mediterranean sea. All of the procedures in this study are not novel. It is just a duplication of lab process to another animals. However, it has some important meaning for the biological data collection for this species. To be honest, the writing of this paper is not good, need a lot of hard working. And the discussion part should be deeply improved in the further version.
Here are some minor comments:
Line 50: This sentence may lack a subject. It should be: “It suggested” or “This suggested…”. [12] should go back before full stop.
Line 52: The same mistake as the Line 50, lack of subject. And [13] should go back before full stop.
Line 57: Usually, the “stylets” is a kind of hard structure in octopus, and “gladius” should be the same kind of hard structure in squid, rather than in octopus.
Line 79: It is much better to use a consistent expression. “Upper beak” and “Lower beak”, instead of “upper jaw and lower jaw”.
Line 90: [13] with a comma??
Figure 2, 4 and 5: The formulation in the figure. The comma should be dot.
Author Response
Comment 1: This paper was described the aging of Sepia officinalis in the Mediterranean Sea. All of the procedures in this study are not novel. It is just a duplication of lab process to another animals. However, it has some important meaning for the biological data collection for this species. To be honest, the writing of this paper is not good, need a lot of hard working. And the discussion part should be deeply improved in the further version.
Response 1: Thank you very much for your comments and suggestions.
We believe that the work presents some novelties:
- We performed for the first time age analysis in S.officinalis in the Mediterranean Sea using beaks and statoliths on wild specimens at different sizes and maturity stages.
- Growth increments in the beaks were counted in the lateral wall and not in the RSS (more common).
- Beaks were cross-checked by comparing their counts with those of statoliths (hard structure validated in this species) extracted from the same specimens.
We confirm that the methodologies are not new; they are the same used by all researchers because they have been validated so far.
We do not know how to improve the discussion without falling into speculation but following the suggestions of all the reviewers we have tried to implement it.
Comment 2: Line 50: This sentence may lack a subject. It should be: “It suggested” or “This suggested…”. [12] should go back before full stop.
Response 2: We agree. The subject was the authors of the reference number 12. After your comment we have modified the sentence
Comment 3: Line 52: The same mistake as the Line 50, lack of subject. And [13] should go back before full stop.
Response 3: We agree. Also in this case the subject was the authors of the reference number 13. After your comment we have modified the sentence
Comment 4: Line 57: Usually, the “stylets” is a kind of hard structure in octopus, and “gladius” should be the same kind of hard structure in squid, rather than in octopus.
Response 4: The sentence did not refer to octopus but was a general consideration on other hard structures in addition to the statoliths present in cephalopods and used for age estimation
Comment 5: Line 79: It is much better to use a consistent expression. “Upper beak” and “Lower beak”, instead of “upper jaw and lower jaw”.
Response 5: We have followed the suggestion replacing the expression jaw whith beak
Comment 6: Line 90: [13] with a comma??
Response 6: We have modified the sentence.
Comment 7: Figure 2, 4 and 5: The formulation in the figure. The comma should be dot.
Response 7: Thank you. We have replaced all commas with dots in the figures, tables and along the manuscript
Reviewer 2 Report
Comments and Suggestions for Authors
I am happy to see new research on cephalopod age and growth using hard structures. The paper has scientific value and increases the understanding of the lifespan of S. officinalis which is an important fisheries resource in the study region. I think that the authors could also include some growth rate estimates in the study.
You will find my suggestions to improve the manuscript attached.

The writing could be improved, I suggest that the language could be reviewed (maybe again?) by a native speaker.
Author Response
Comment 1: I am happy to see new research on cephalopod age and growth using hard structures. The paper has scientific value and increases the understanding of the lifespan of S. officinalis which is an important fisheries resource in the study region. I think that the authors could also include some growth rate estimates in the study. You will find my suggestions to improve the manuscript attached.
Response 1: Thank you very much for your comments and suggestions (attached in the manuscript) that have been addressed in the new version. We have tried to improve the image quality and size of the legend of figures 2 and 4 and have modified the legend of figure 3 as you suggested. We have followed your suggestion to calculate growth increment. Some details (e.g. the damaged areas in the beak) that were not clear enough in the original manuscript, now are better explained both in the results and in the discussion of the new version. Thank you also for the suggested references that had been useful.
Comment 2: The writing could be improved, I suggest that the language could be reviewed (maybe again?) by a native speaker.
Response 2: The manuscript has been already checked by a mother tongue before to be submitted. We reserve to check again the English language, eventually, in an onother step.
To answer to some questions present in your revised pdf
Question1: As regard the sentence of the introduction (This species together with Octopus vulgaris, 39
Eledone moschata and E. cirrhosa, accounts for 15% of the total landing value [8].)
For most of the Mediterranean areas or specifically in the Strait of Sicily??
Response 1: the value is referred to the most of the Mediterranean areas.
Question 2: Was the experimental project related to keep
Response 2: The main objective of the experimental project was to assess the reproductive aspect of S.officinalis in Sardinian waters by means of fishing (traps and nets), with the ultimate aim of managing and protecting the reproductive phase. The data reported in this work were extrapolated from the analysed samples.
Question 3: image analysis software? Image Pro? ImageJ? I think it is nice if you specify which one since some of them have procedures to count growth mrks in trees and otoliths...
Response 3: We counted increments by an image analysis using Tps_Dig2 software (see new Material and Methods)
Question 4: Did you use imersion oil when reading it? some low density imersion oil and increasing the contrast by closing the light apperture. Also when using and image analysis system, change to black and white, so you remove the yellowish nature of the images and increase the contrast. Nice to see the hatching "natal ring" in this image, but it is out of focus so we can not count how may days it took to hatch...
Response 4: We didn't use imersion oil to read increments. We know the useful of postproduction images changing them (e.g. in black and withe) but we have preferred to furnish the original image that clearly show the readiblity of statoliths increments from the nucleus to the end of the border without any modification of the image.
Question 5: An artifact that could explain it is that near to the dome border increment width is reduced, so you need to keep changing the focus as you move to the border, what it is impossible if you keep the same focus from nucleus to the border. It is one thing to count twice the statolith "images" taken from the statolith, and another is to "read" the statolith on the microscope with a hand counter twice in 15 days interval. With the beaks is easier since it is a 2D structure and increments are oreders of magnitude wider. The same thing happens wth statoliths and the gladius.
Response 5: We agree that statoliths are more complex and difficult to read than beaks. For this reason we conclude our manuscript by suggesting the beak as a hard structure suitable for estimating age in S.officinalis. We think that reading the statolith under the microscope with one hand counter lead to more errors than reading by images, especially when the counting has being done by one reader as in our case. However, the counting was done by a trained reader that knows the importance to change the focus from core to boundary.
Question 6: But mature or near to death individuals have lower growth rates since they reduce metabolic rates and feeding activity, which could result in an increased increment deposition times..
Response 6: This consideration is interesting and suggests the need for future studies addressing metabolic and age investigations not only in Sepia officinalis but also in other coleoid species. However, this conclusion could questioned the daily periodicity in the adult phase validated for some squid species.....
Reviewer 3 Report
Comments and Suggestions for Authors
I have reviewed the ms “Age estimation in Sepia officinalis using beaks and statoliths” and I find it interesting due to current interest in cephalopod fisheries and biology. However the ms has two main flaws. There is no validation for growth increments analysed in the ms, so conclusions are very limiting. Also growth increment counting was done by the same operator, which compromise the reability of the results.
Further comments are enlisted below:
Lines 56-58, “Other hard structures, such as stylets gladius and eye lens have also been investigated to determine their usability as an ageing tool [21, 22, 23 ].” Here a pivotal reference is lacking, Rodríguez-Domínguez et al. (2013) which compared growth increments from beaks, stylets and eye lenses.
Line 97, “…the counting was done twice by the same operator…”, the counting should have been done by at least two different readers, see Arkhipkin et al. (2018), Campana (2001).
Line 233-241, the damaged areas in the beak lateral wall has nothing to do with the erosion of the rostrum tip, so the problem of underestimating age in large individuals is still present.
Line 231-243, poor preparation of statoliths could be due also because using dead animals (line 73) in which acidic decomposition may damage statoliths beyond readability.
Table 1 and 2, better use N than N°, i.e. N=72.
Rodríguez-Domínguez A, Rosas C, Méndez-Loeza I, Markaida U (2013) Validation of growth increments in stylets, beaks and lenses as ageing tools in Octopus maya. Jour Exp Mar Biol Ecol 449:194–199.
Author Response
Comment 1: I have reviewed the ms “Age estimation in Sepia officinalis using beaks and statoliths” and I find it interesting due to current interest in cephalopod fisheries and biology. However the ms has two main flaws. There is no validation for growth increments analysed in the ms, so conclusions are very limiting. Also growth increment counting was done by the same operator, which compromise the reability of the results.
Response 1: The present paper aims to evaluate the age estimation in specimens of Sepia officinalis caught in the Mediterranean Sea using statoliths and beaks to test the reliability of these structures and to suggest the best one suitable for age study in this species. There is no validation for growth increments but beaks were cross-checked by comparing their counts with those of statoliths extracted from the same specimens. As regard the counting done we explained below in the specific comment
Comment 2: Lines 56-58, “Other hard structures, such as stylets gladius and eye lens have also been investigated to determine their usability as an ageing tool [21, 22, 23 ].” Here a pivotal reference is lacking, Rodríguez-Domínguez et al. (2013) which compared growth increments from beaks, stylets and eye lenses.
Response 2: Thank you. Now we have inserted this reference
Comment 3: Line 97, “…the counting was done twice by the same operator…”, the counting should have been done by at least two different readers, see Arkhipkin et al. (2018), Campana (2001).
Response 3: Thanks for this warning. We are familiar with the methodology of Arkhipkin et al. (2018), however we confirm that our countings were done by one reader at different times but we would stressed that she was a trained reader (first author). To be honest we don't think that this methodology would compromised the results. At the beginning of the analysis we didn’t exclude the possibility to do a third count if the increment number differed by more than 10%, (we didn’t write this in the material and methods becuase it didn’t happen. We would also highlighted that other scientific papers focusing on age and performed on beaks and statoliths report the same our procedure. To list some: Schwarz et al (2019); Guerra Marrero et al 2023a,b; Perales Raya et al., 2010; Bettencourt and Guerra, 2001
Comment 4: Line 233-241, the damaged areas in the beak lateral wall has nothing to do with the erosion of the rostrum tip, so the problem of underestimating age in large individuals is still present.
Response 4:Thank you. We agree. The sentence was not clear. Now we have tried to better explain the sentence about the damaged areas (scratches) observed in the lateral wall of the beaks of few bigger animals.
Comment 5: Line 231-243, poor preparation of statoliths could be due also because using dead animals (line 73) in which acidic decomposition may damage statoliths beyond readability.
Response 5: Thank you, but we prefer not to include this hipotesis because our samples, although dead, after being collected were immediately analysed and their statoliths extracted for processing.
Comment 6:Table 1 and 2, better use N than N°, i.e. N=72.
Response 6: Than you, now we have modified them
Round 2
Reviewer 1 Report
Comments and Suggestions for Authors
I agree to publish this paper as the current form.
Author Response
Comment: I agree to publish this paper as the current form.
Response: Thank you for your agreement.
Reviewer 3 Report
Comments and Suggestions for Authors
I found the ms notably improved after the revision.
Lines 129-130. “in some peripheral regions of the lateral wall of a few beaks belonging to the largest individuals “ This sentence has to be rewritten. Also, give the number of such individuals better than “a few”.
My only last comment is that although sexual maturity was determined, no information on size or age at maturity are given in Results, which would make the ms more interesting.
Sexual maturity is considered in Discussion (lines 288-290), but this should have been described in Results in some detail.
Line 325, “It is also known the existence of two biological cycles one in the Mediterranean…”. Better change to “The biological cycle in the Mediterranean lasts 1-year and lay eggs at smaller mature sizes, while in the Atlantic…”
Author Response
Comment 1: I found the ms notably improved after the revision.
Response1: Thank you for your agreement.
Comment 2: Lines 129-130. “in some peripheral regions of the lateral wall of a few beaks belonging to the largest individuals “This sentence has to be rewritten. Also, give the number of such individuals better than “a few”.
Response 2: Now, we have inserted the number of the largest individuals as you requested (see new line 131)
Comment 3: My only last comment is that although sexual maturity was determined, no information on size or age at maturity are given in Results, which would make the ms more interesting.
Response 3: The specimens were collected for age analysis and prior to this were sexed and assessed in their maturity level. We believe that the available data are not suitable for developing maturity age curves. Therefore, to follow your comment, we can only include in the text some details on the minimum size at maturity and the average age of our mature specimens for both sexes as a further explanation of Table 2 (new lines 125-26; 162-164).
Comment 4: Sexual maturity is considered in Discussion (lines 288-290), but this should have been described in Results in some detail.
Response 4: Now, new details on sexual maturity are included in the results (new lines 125-26; 162-164).
Comment 5: Line 325, “It is also known the existence of two biological cycles one in the Mediterranean…”. Better change to “The biological cycle in the Mediterranean lasts 1-year and lay eggs at smaller mature sizes, while in the Atlantic…”
Response 5: As you requested, we have modified the sentence (see new lines 162-164).